# Neutrophil-to-Lymphocyte and Platelet-to-Lymphocyte Ratios as Prognostic Biomarkers in Unresectable Hepatocellular Carcinoma Treated with Atezolizumab plus Bevacizumab

**DOI:** 10.3390/cancers14235834

**Published:** 2022-11-26

**Authors:** Yue Linda Wu, Claudia Angela Maria Fulgenzi, Antonio D’Alessio, Jaekyung Cheon, Naoshi Nishida, Anwaar Saeed, Brooke Wietharn, Antonella Cammarota, Tiziana Pressiani, Nicola Personeni, Matthias Pinter, Bernhard Scheiner, Lorenz Balcar, Yi-Hsiang Huang, Samuel Phen, Abdul Rafeh Naqash, Caterina Vivaldi, Francesca Salani, Gianluca Masi, Dominik Bettinger, Arndt Vogel, Martin Schönlein, Johann von Felden, Kornelius Schulze, Henning Wege, Peter R. Galle, Masatoshi Kudo, Lorenza Rimassa, Amit G. Singal, Rohini Sharma, Alessio Cortellini, Vincent E. Gaillard, Hong Jae Chon, David J. Pinato, Celina Ang

**Affiliations:** 1Department of Medicine, Division of Hematology/Oncology, Tisch Cancer Institute, Icahn School of Medicine at Mount Sinai, New York, NY 10029, USA; 2Department of Surgery & Cancer, Imperial College London, Hammersmith Hospital, Du Cane Road, London W12 0HS, UK; 3Division of Medical Oncology, Policlinico Universitario Campus Bio-Medico, 00128 Rome, Italy; 4Department of Biomedical Sciences, Humanitas University, Via Rita Levi Montalcini 4, Pieve Emanuele, 20072 Milan, Italy; 5Medical Oncology, Department of Internal Medicine, CHA Bundang Medical Center, CHA University, Seongnam 46371, Republic of Korea; 6Department of Gastroenterology and Hepatology, Faculty of Medicine, Kindai University, Osaka 589-8511, Japan; 7Department of Medicine, Division of Medical Oncology, Kansas University Cancer Center, Kansas City, KS 66160, USA; 8Medical Oncology and Hematology Unit, Humanitas Cancer Center, IRCCS Humanitas Research Hospital, Via Manzoni 56, Rozzano, 20089 Milan, Italy; 9Division of Gastroenterology and Hepatology, Department of Internal Medicine III, Medical University of Vienna, 1090 Vienna, Austria; 10Division of Gastroenterology and Hepatology, Department of Medicine, Taipei Veterans General Hospital, Taipei 11217, Taiwan; 11Institute of Clinical Medicine, National Yang Ming Chiao Tung University, Taipei 71150, Taiwan; 12School of Medicine, National Yang Ming Chiao Tung University, Taipei 11217, Taiwan; 13Department of Internal Medicine, University of Texas Southwestern Medical Center, Dallas, TX 75390, USA; 14Medical Oncology/TSET Phase 1 Program, Stephenson Cancer Center, University of Oklahoma, Oklahoma City, OK 73104, USA; 15Department of Translational Research and New Technologies in Medicine and Surgery, University of Pisa, 56124 Pisa, Italy; 16Unit of Medical Oncology 2, Azienda Ospedaliero-Universitaria Pisana, 56126 Pisa, Italy; 17Sant’Anna School of Advanced Studies, 56127 Pisa, Italy; 18Department of Medicine II (Gastroenterology, Hepatology, Endocrinology and Infectious Diseases), Freiburg University Medical Center, Faculty of Medicine, University of Freiburg, 79106 Freiburg, Germany; 19Department of Gastroenterology, Hepatology and Endocrinology, Hannover Medical School, 30625 Hannover, Germany; 20Department of Oncology, Hematology and Bone Marrow Transplantation with Section of Pneumology, University Medical Center Hamburg-Eppendorf, 20251 Hamburg, Germany; 21I. Department of Medicine, University Medical Center Hamburg-Eppendorf, 20251 Hamburg, Germany; 22I. Department of Internal Medicine, University Medical Center Mainz, 55131 Mainz, Germany; 23F. Hoffmann-La Roche Ltd., 4070 Basel, Switzerland

**Keywords:** NLR, PLR, hepatocellular carcinoma, atezolizumab plus bevacizumab, immunotherapy, inflammatory markers

## Abstract

**Simple Summary:**

Immunotherapy is now the standard front-line therapy for patients with advanced hepatocellular carcinoma. However, there remains a substantial proportion of patient who do not respond to this treatment, and few predictive and prognostic biomarkers exist that can identify patients most likely to benefit from immunotherapy. Inflammation plays a role in driving tumor formation and progression. The aim of our study was to evaluate the prognostic utility of two blood-based markers of inflammation, which have the advantage of being easily accessible and inexpensive, and we found that one may predict survival outcomes in patients with hepatocellular carcinoma treated with our current standard of care immunotherapy regimen.

**Abstract:**

Systemic inflammation is a key risk factor for hepatocellular carcinoma (HCC) progression and poor outcomes. Inflammatory markers such as the neutrophil-to-lymphocyte ratio (NLR) and platelet-to-lymphocyte ratio (PLR) may have prognostic value in HCC treated with standard of care atezolizumab plus bevacizumab (Atezo-Bev). We conducted a multicenter, international retrospective cohort study of patients with unresectable HCC treated with Atezo-Bev to assess the association of NLR and PLR with overall survival (OS), progression-free survival (PFS), and objective response rates. Patients with NLR ≥ 5 had a significantly shorter OS (9.38 vs. 16.79 months, *p* < 0.001) and PFS (4.90 vs. 7.58 months, *p* = 0.03) compared to patients with NLR < 5. NLR ≥ 5 was an independent prognosticator of worse OS (HR 2.01, 95% CI 1.22–3.56, *p* = 0.007) but not PFS. PLR ≥ 300 was also significantly associated with decreased OS (9.38 vs. 15.72 months, *p* = 0.007) and PFS (3.45 vs. 7.11 months, *p* = 0.04) compared to PLR < 300, but it was not an independent prognosticator of OS or PFS. NLR and PLR were not associated with objective response or disease control rates. NLR ≥ 5 independently prognosticated worse survival outcomes and is worthy of further study and validation.

## 1. Introduction

Hepatocellular carcinoma (HCC) is a leading cause of cancer-related mortality worldwide [1], and until recently, systemic therapy for patients with unresectable or advanced disease consisted of sorafenib and other multikinase inhibitors, which added limited efficacy but high toxicity [2,3]. However, immune checkpoint inhibitors (ICIs) have now revolutionized the treatment landscape for unresectable HCC. In particular, the IMbrave150 trial found that the combination of the programmed-death ligand-1 (PD-L1) inhibitor, atezolizumab, and vascular endothelial growth factor (VEGF) inhibitor, bevacizumab, improved overall survival (OS) compared to sorafenib [4,5]. This combination has since become the standard frontline therapy for patients with unresectable HCC. Nevertheless, approximately 20–25% of patients experience primary resistance to atezolizumab plus bevacizumab (Atezo-Bev) [5], highlighting the need to identify those who may derive the most benefit from this therapy.

To date, no standard biomarkers exist in HCC that effectively predict response or resistance to ICI therapy. While PD-L1 expression is a validated predictive biomarker of ICI response in lung and urothelial carcinomas, it has not been consistently correlated with response to ICI in HCC in clinical trials using nivolumab [6], pembrolizumab [7], or Atezo-Bev [8]. Both tumor mutational burden and microsatellite instability have also been associated with ICI response in other solid tumors, but their low prevalence in HCC limits their utility as predictive biomarkers [9,10]. As the HCC treatment paradigm shifts to immunotherapy, discovery of potentially predictive and prognostic biomarkers has gained interest.

Local and systemic inflammation have been shown to be key drivers of HCC tumorigenesis and progression, as the pro-inflammatory state of chronic liver disease leads to progression to fibrosis and malignancy [11]. In patients treated with ICIs, therapeutic response has been linked to higher CD3+ and CD8+ T cells infiltration [12,13], high effector T cell expression [14], and activation of the IFNγ signaling pathway [13,15]. Conversely, activating mutations in the WNT/β-catenin signaling pathway and increased regulatory T cell expression have been found to contribute to immunotherapy resistance [14,16,17]. While these findings have provided invaluable insight into the complex interplay of antitumor effector cells and their suppressors in the tumor microenvironment, ultimately dictating response or resistance to ICIs, these biomarkers cannot be widely applied in the clinical setting due to prohibitive cost and the need for biopsy specimens that are often unavailable as diagnosis of HCC can be made based on radiographic characteristics alone [18].

Systemic inflammation has been linked with worse clinical outcomes in many cancer types [19]. Two such markers of systemic inflammation include the neutrophil-to-lymphocyte ratio (NLR) and the platelet-to-lymphocyte ratio (PLR), which are readily available in routine clinical practice and inexpensive. In HCC, an elevated NLR may be indicative of a pro-inflammatory tumor environment [20] and has been shown to correlate with worse survival outcomes [21,22]. For patients with HCC treated with ICI before the widespread adoption of Atezo-Bev, both elevated NLR and PLR were found to be independent prognostic factors for worse survival outcomes [23,24]. The aim of the current study was to better understand the prognostic value of NLR and PLR in a real-world cohort of patients with unresectable HCC treated with Atezo-Bev, which has become the contemporary standard of care therapy.

## 2. Materials and Methods

### 2.1. Patient Population

The study population consisted of patients who received Atezo-Bev for unresectable HCC at 14 institutions across the United States, Europe, and Asia between January 2019 and April 2022. Patients included in the analysis had a radiologic or histologic diagnosis of HCC in accordance with American Association for the Study of Liver Disease [25] and European Association for the Study of the Liver [26] guidelines. In addition, patients were required to have unresectable HCC defined as Barcelona Clinic Liver Cancer (BCLC) stage B unsuitable for locoregional therapies or stage C, Child Pugh (CP) class A liver function, Eastern Cooperative Oncology Group (ECOG) performance status 0 or 1, and treatment with Atezo-Bev in the first line, all of which are consistent with the IMbrave150 trial inclusion criteria [4]. Baseline esophagogastroduodenoscopies were performed at the discretion of each institution. Of the 433 total patients in the prospectively maintained database, 296 patients met the criteria for inclusion in the analysis.

### 2.2. Study Design

Patient demographics and clinical data, including BCLC stage, CP class, ECOG performance status, alpha fetoprotein (AFP) level, presence of cirrhosis (clinically or radiologically diagnosed), presence of extrahepatic metastases, presence of neoplastic portal venous thrombosis (PVT), etiology of liver disease, prior locoregional therapy, follow-up, and vital status, were collected retrospectively. Baseline data were defined as the time of Atezo-Bev initiation. Treatment response was evaluated using computerized tomography and/or magnetic resonance imaging approximately every 9 weeks during treatment, and responses were determined by RECIST 1.1 criteria based on local institutional assessment.

The primary objective of the study was to determine whether there was an association between NLR and OS and between PLR and OS. Secondary objectives included assessing the association of NLR and PLR with progression-free survival (PFS), and the effect of NLR and PLR on objective response rate (ORR), defined as the proportion of patients achieving a radiographic complete response (CR) or partial response (PR) by RECIST 1.1 criteria. The effect of NLR and PLR on the disease control rate (DCR) was also investigated, and DCR was defined as the proportion of patients achieving a radiographic CR, PR, or stable disease (SD) according to RECIST 1.1 criteria.

NLR was calculated as the ratio of the total neutrophil count to the absolute lymphocyte count (ALC), and PLR was calculated as the ratio of platelet count to ALC. These ratios were obtained at baseline, prior to the initiation of Atezo-Bev therapy. In previous studies of NLR and PLR in HCC, NLR ≥ 5 and PLR ≥ 300 have been used and found to have negative prognostic significance [23,24]. In this study, these same cutoff values were used to establish NLR and PLR groups. Finally, the effect of NLR and PLR status on the incidence of adverse events (AEs) was also evaluated. AEs were defined and graded based on the Common Terminology Criteria for Adverse Events (CTCAE) classification, version 5.0, and identified based on investigator review of clinical notes, radiographic, and laboratory data.

### 2.3. Statistical Analysis

Patient demographic and clinicopathological characteristics were reported descriptively as medians and interquartile ranges for continuous variables and percentages for qualitative ones. OS was defined as the time from treatment start to death from any cause. Patients still alive at the time of data cut-off were censored at the last follow-up. PFS was calculated as the time from treatment commencement to death or radiological progression, whichever came first. Patients not reporting progression at the time of data cut-off were censored at the time of last follow-up. Median OS and PFS were estimated using Kaplan–Meier method, whereas median follow-up times were estimated with the reverse Kaplan–Meier method. Fisher’s exact test was performed to determine associations between categorical variables while the Mann–Whitney test was used to compare continuous variables, and a *p* value < 0.05 was considered to represent a significant association. After testing for proportionality of the hazards assumption, univariable and multivariable Cox proportional hazard models were used to evaluate the prognostic impact of baseline clinico-pathologic characteristics on OS and PFS, to obtain hazard ratios (HR) and 95% CI. NLR and PLR were included in the model as categoric variables with a cut-off of ≥5 and ≥300, respectively. Covariates were selected for the multivariable models if they were found to be significant in univariable analysis. To overcome the risk of bias associated with multicollinearity in the multivariate model including both NLR and PLR, we constructed two multivariate models and tested the fitness of each model by performing likelihood ratio tests. Fisher’s exact tests were performed to determine the relationship between inflammatory marker groups and measures of response (ORR and DCR) and to examine the association of these markers and the incidence of AEs. All statistical analyses were carried out using R studio version 2021.09.2 and IBM SPSS statistics version 26.

## 3. Results

### 3.1. Baseline Characteristics

Baseline patient characteristics are reported in Table 1. A total of 296 patients met the inclusion criteria. Among the 137 excluded patients, 94 did not meet the inclusion criteria due to deranged liver function (CP class B or C), 39 received Atezo-Bev beyond first line, and 4 had an ECOG PS at baseline higher than 1. The median age of the cohort was 66 years (interquartile range [IQR]: 59–73), and the majority of patients were male (*N* = 245, 82.7%) and had underlying cirrhosis (*N* = 222, 75.0%), with most patients having a viral etiology of liver disease (*N* = 195, 65.9%). At the time of Atezo-Bev initiation, 204 patients (68.9%) had BCLC stage C disease and the remainder (*N* = 169, 51.7%) had BCLC stage B disease. One hundred and sixty-nine patients (51.7%) had extrahepatic spread and 104 (35.0%) had evidence of neoplastic PVT. All patients included in this cohort had preserved liver function (CP score 5: *N* = 190, 64.2%; CP score 6: *N* = 106, 35.8%) and good performance status (ECOG 0: *N* = 139, 47.0%; ECOG 1: *N* = 157, 53.0%). There were 161 patients (54.4%) with albumin-bilirubin (ALBI) grade 1 and 135 patients (45.6%) with ALBI grade 2 at the start of therapy, and patients had a median AFP of 70 ng/mL (IQR: 6.5–1525). Most patients had been previously treated with locoregional therapy (*N* = 186, 59.6%).

Out of the 296 patients included in the analysis, 15 patients were missing NLR and PLR data. For the evaluable cohort, the median NLR was 2.89 (IQR: 1.80–4.47), and the median PLR was 139 (IQR: 93–196). The high NLR group (NLR ≥ 5) consisted of 56 patients, and the remaining 225 patients were in the low NLR group (NLR < 5). Patients in the NLR ≥ 5 group tended to have a higher median AFP level (268 vs. 62, *p* = 0.02), incidence of PVT (51.9% vs. 32.9%, *p* = 0.02), more advanced liver disease, including a greater number of patients with CP score 6 (48.1% vs. 32%, *p* = 0.03) and ALBI grade 2 (67.9% vs. 39.9%, *p* < 0.001). Other characteristics were not significantly different compared to patients with NLR < 5. There were 24 patients in the PLR ≥ 300 group and 257 patients in the PLR < 300. Median platelet count was 154 for patients with PLR ≥ 300 (IQR: 113–217) and 155 (IQR: 111–218) for patients with PLR < 300. There were no statistically significant differences in baseline patient characteristics when stratified by PLR level.

### 3.2. Survival Outcomes

The cohort was followed for a median of 9.93 months (95% CI 9.4–10.5). The median OS of patients with NLR ≥ 5 was 9.38 months (95% CI 6.94-Not Reached [NR]) and 16.79 months for patients with NLR < 5 (95% CI 14.68-NR) (Figure 1A). Similarly, for patients with PLR ≥ 300, the median OS was 9.38 (95% CI 6.45-NR), and for patients with PLR < 300 (95% CI 14.48-NR), the median OS was 15.72 months (Figure 1B). In univariate analysis, both NLR ≥ 5 and PLR ≥ 300 were significantly associated with worse OS (HR 2.71, 95% CI 1.71–4.27, *p* < 0.001; HR 2.24, 95% CI 1.71–4.27, *p* = 0.007). In multivariate analysis, NLR ≥ 5 remained an independent prognosticator of worse OS with Atezo-Bev (HR 2.01, 95% CI 1.22–3.56, *p* = 0.007) whereas PLR level was not (HR 1.01, 95% CI 0.52–1.96, *p* = 0.99) (Table 2). These results are consistent when accounting for possible collinearity (Appendix A).

In univariable analyses, other predictors of worse OS with Atezo-Bev included AFP ≥ 400 (HR 1.72, 95% CI 1.15–2.59, *p* = 0.009), ALBI grade 2 (HR 3.65, 95% CI 2.36–5.64, *p* < 0.001), CP score 6 (HR 2.42, 95% CI 1.24–4.05, *p* < 0.001), presence of neoplastic PVT (HR 2.03, 95% CI 1.39–2.99, *p* < 0.001), and prior treatment with locoregional therapy (HR 0.52, 95% CI 0.35–0.79, *p* = 0.002) (Table 2). However, in multivariable analyses, only ALBI grade 2 remained an independent prognosticator of shorter OS compared to ALBI grade 1 (HR 2.35, 95% CI 1.42–3.89, *p* < 0.001). Variables such as age, gender, body mass index (BMI), BCLC stage, presence of cirrhosis, extrahepatic spread, performance status, and viral vs. non-viral etiology of underlying liver disease were not found to be significantly associated with OS.

Next, the effect of these systemic inflammatory markers on PFS was investigated and results are shown in Table 3. While univariate analysis showed that NLR ≥ 5 was associated with worse PFS (HR 1.54, 95% CI 1.05–2.25, *p* = 0.03), no significant association with PFS was found in multivariate analysis (HR 1.31, 95% CI 0.84–2.04, *p* = 0.24). The median PFS of patients with NLR ≥ 5 was 4.90 months (95% CI 3.45–7.17) and 7.58 months (95% CI 6.41–9.44) for patients with NLR < 5 (Figure 2A). Similarly, PLR ≥ 300 was correlated with a significantly worse PFS (HR 1.72, 95% CI 1.04–2.83, *p* = 0.04) in univariate analysis that failed to be confirmed in multivariate analysis (HR 1.18, 95% CI 0.65–2.13, *p* = 0.59). Again, these results are unchanged when accounting for potential collinearity (Appendix A). Patients with PLR ≥ 300 had a median PFS of 3.45 months (95% CI 1.81–7.14) whereas patients with PLR < 300 had a median PFS of 7.11 months (95% CI 6.22–8.29) (Figure 2B).

AFP ≥ 400 (HR 1.51, 95% CI 1.11–2.05, *p* = 0.009) and ALBI grade 2 (HR 1.57, 95% CI 1.16–2.2, *p* = 0.003) were associated with worse PFS in univariate analysis and remained independent prognosticators of shorter PFS in multivariate analysis (HR 1.41, 95% CI 1.02–1.93, *p* = 0.035; HR 1.40, 95% CI 1.03–1.92, *p* = 0.034) (Table 3). Other variables including age, gender, BMI, BCLC stage, CP score, presence of cirrhosis, presence of neoplastic PVT, etiology of HCC, and prior locoregional therapy were not associated with PFS (Table 3).

### 3.3. Response to Atezolizumab plus Bevacizumab

Patients with known baseline NLR and PLR levels were evaluated for response to Atezo-Bev therapy. There was no significant difference in ORR between patients who had NLR ≥ 5 (24%) compared to patients with NLR < 5 (32%, *p* = 0.39) (Figure 3A). No significant difference in ORR was seen between patients with PLR ≥ 300 and PLR < 300 (33% vs. 30%, *p* = 0.81) (Figure 3B). Similarly, there was no difference in DCR when comparing NLR ≥ 5 vs. NLR < 5 (71% vs. 79%, *p* = 0.24), and patients with PLR ≥ 300 did not have a significantly different DCR compared to patients with PLR < 300 (62% vs. 79%, *p* = 0.09) (Figure 3C,D).

### 3.4. Adverse Events

Finally, the incidence of AEs with Atezo-Bev and the association with NLR and PLR were assessed in this population, and results are shown in Table 4. A total of 221 patients (74.7%) experienced an AE of any grade, with 70 patients (23.6%) experiencing a grade 3 or higher treatment-related AE. In addition, 63 patients (21.3%) developed immune-related AEs with this combination therapy. When stratified by NLR ≥ 5 and NLR < 5, the rate of treatment-related AEs of any grade was not statistically different (66.1% vs. 77.8%, *p* = 0.08), but patients with NLR < 5 were at higher risk of developing grade 3 or higher AEs compared to patients with NLR ≥ 5 (27.5% vs. 14.8%, *p* = 0.04). The PLR levels, ≥300 or <300, were not significantly associated with increased or decreased incidence of developing treatment-related AEs, though patients with PLR < 300 appeared to have a borderline statistically significant increased risk of developing an AE of any grade (77.1% vs. 58.8%, *p* = 0.05). There was no difference in the incidence of immune-related AEs between either NLR ≥ 5 vs. NLR < 5 groups (22.2% vs. 21.8%, *p* = 0.96) or between PLR ≥ 300 vs. PLR < 300 groups (20.8% vs. 21.8%, *p* = 0.91).

### 3.5. Discontinuation of Therapy

Patients in this cohort were treated for a median duration of 7.30 months (95% CI 6.3–8.7), with 97 patients (32.8%) still receiving therapy at the time of data cutoff. Of note, however, patients with NLR ≥ 5 were only treated for a median duration of 3.53 months (95% CI 3.0–8.1), with 15 out of the 56 patients (26.8%) still receiving therapy at the cutoff date. Conversely, patients with NLR < 5 received Atezo-Bev for a median duration of 8.39 months (95% CI 6.6–10.2), with 72 out of 225 patients (32.0%) still receiving therapy at the time of analysis. Of the 41 patients with NLR ≥ 5 who stopped therapy, 7 (17.0%) patients died on therapy, 17 (41.4%) had progressive disease (PD), 9 (22.0%) had treatment-related AEs, and 7 (17.0%) had clinical deterioration unrelated to AEs. Of the 153 patients with NLR < 5 who discontinued therapy, 5 (3.2%) patients died on therapy, 104 (68.0%) had PD, 15 (9.8%) had treatment-related AEs, and 15 (9.8%) had clinical deterioration.

Patients with PLR ≥ 300 were treated with Atezo-Bev for a median duration of 6.88 months (95% CI 3.9–14.2) while patients with PLR < 300 were treated for a median duration of 7.70 months (95% CI 6.3–9.1). At the time of analysis, 5 patients out of 24 (20.8%) with PLR ≥ 300 and 82 patients out of 257 (31.9%) with PLR < 300 remained on therapy. Of the 19 patients with PLR ≥ 300 who discontinued Atezo-Bev, 3 patients (15.8%) died on therapy, 10 (52.6%) had PD, 3 (15.8%) had treatment-related AEs, and 3 (15.8%) had clinical deterioration. Of the 175 patients with PLR < 300 who discontinued therapy, 6 patients (3.4%) had died on therapy, 111 (63.4%) had PD, 21 (12.0%) had treatment-related AEs, and 19 (10.9%) had clinical deterioration.

## 4. Discussion

Our multi-center, international, retrospective cohort study examined the prognostic value of NLR and PLR in patients with unresectable HCC treated with the current standard front-line therapy, Atezo-Bev. Given the clinically relevant proportion of patients without objective response to Atezo-Bev and lack of practical biomarkers that can be used to predict response to ICIs, we aimed to evaluate the prognostic utility of NLR and PLR, which are inexpensive and easily accessible, in patients with unresectable HCC treated with Atezo-Bev. We found that while high NLR, defined as NLR ≥ 5, and high PLR, defined as PLR ≥ 300, were associated with worse OS, only high NLR was an independent prognosticator of worse OS in multivariate analyses. In addition, in univariate analyses, both high NLR and PLR were correlated with worse PFS, but neither variable was independently prognostic of PFS. Our results also suggested that baseline NLR and PLR did not correlate with objective response or disease control.

Findings from this study contribute to an emerging body of evidence elucidating the clinical value of systemic markers of inflammation in predicting outcomes with immunotherapy in unresectable HCC. These results are consistent with a prior study showing that NLR < 5 was associated with prolonged OS and PFS compared to NLR ≥ 5 in patients with HCC treated with nivolumab [23]. In the same study, when PLR was divided in terciles, higher PLR was associated with worse OS but not PFS, and neither NLR nor PLR was correlated with response to nivolumab [23]. In a previous study conducted by our group evaluating NLR and PLR in patients with HCC treated with a variety of ICIs but not including Atezo-Bev, NLR ≥ 5 was found to be an independent negative prognosticator of OS, and PLR ≥ 300 independently predicted worse OS and PFS [24]. In our current study, PLR ≥ 300 was not an independent prognostic marker, but this may have been due to the small sample size (*N* = 24). Again, in Muhammed et al., neither variable was an independent predictor of response to immunotherapy [23]. Our data are also in agreement with a recent Japanese study consisting of 249 patients that also investigated NLR in unresectable HCC treated with Atezo-Bev and found that high NLR, with a cutoff of ≥3, was associated with worse OS (HR 3.37, 95% CI 1.02–11.08) but not with response [27]. However, ours is the only study to our knowledge that evaluated NLR and PLR in a cohort of patients across three continents with HCC treated with Atezo-Bev.

The negative prognostic value of high NLR and PLR has been validated in many cancer types with or without therapy with ICIs [28,29,30,31,32] and in HCC treated with hepatic resection, transplantation, locoregional therapy, and tyrosine kinase inhibitors [22,33]. The importance of investigating these inflammatory markers lies not only on their ease of application but also on their role in tumor progression and metastasis. Neutrophilia is associated with immune activation and production of neutrophil-derived cytokines such as VEGF, matrix metalloproteinases, and interleukin-18 (IL-18) that promote inflammation, angiogenesis, and extravasation of leukocytes [19,23]. In addition, relative lymphopenia leads to reduced cell-mediated antitumor immunity due to decreased NK and T cell activity [23,33]. Platelets also release VEGF and platelet-derived growth factor, promoting angiogenesis, cell proliferation, and inflammation [34]. IL-18 impairs NK and T cell function, which in turn impairs recognition of tumor antigens [19,23]. Other proposed mechanisms by which elevated neutrophil count have been reported to promote tumor progression include increased CD163 and IL-17 tumor expression leading to increased peritumoral CD163-positive tumor-associated macrophages which have been shown to play key roles in facilitating metastasis [20], formation of neutrophil extracellular traps consisting of decondensed chromatin with inflammatory proteins that contribute to cancer progression [35], and HCC stem-like cells stimulation [36]. Taken together, a high NLR or PLR may be indicative of disrupted immune modulation of the tumor microenvironment.

High NLR can be linked with greater disease burden and liver dysfunction. In this study, patients with NLR ≥ 5 had higher incidence of neoplastic PVT, more elevated AFP, and a higher CP score and ALBI grade. Nevertheless, our study demonstrated that high NLR was an independent prognosticator of poor OS. However, a higher ALBI grade also independently predicted worse survival in HCC, which is in keeping with previous studies evaluating ALBI as a useful prognostic marker even in patients with CP class A liver disease [37,38]. An AFP level ≥ 400 was independently predictive of shorter PFS in our study but not of OS, but elevated AFP has been well-established as a prognostic biomarker in HCC and correlated with pathologic grade, stage, and tumor size [39]. Because this study is controlled for patients with CP class A liver disease only, these markers can provide insight into the degree of liver dysfunction that CP class alone does not fully capture.

Our study also evaluated the effect of high NLR and PLR on the development of AEs. We found no statistically significant difference in the incidence of treatment-related AEs of any grade between patients with either high or low NLR or PLR, but interestingly, patients with low NLR had a greater rate of grade 3 or higher AEs. This is in contrast to Tada et al., who demonstrated that patients with high NLR were more likely to develop AEs and to discontinue Atezo-Bev due to AEs [27]. Our findings could be partially accounted for by the fact that patients with NLR ≥ 5 were only treated for a median duration of 3.53 months while patients with NLR < 5 were treated for a median of 8.39 months, suggesting that there was insufficient time to observe development of cumulative toxicity. While most patients discontinued therapy due to PD in both groups, more patients with high NLR discontinued Atezo-Bev due to clinical deterioration and death, which could also confound assessment of AEs. Nevertheless, the high mortality and clinical deterioration rates leading to discontinuation of therapy, whether related or unrelated to AEs, of patients with high NLR likely contributes to the shorter OS seen in this group. The high and low PLR groups did not have any statistically significant difference in incidence of AEs, which is likely reflected in the survival outcomes suggesting that PLR may have less prognostic value compared to NLR. Although low NLR and PLR in other solid tumors treated with ICIs were linked with an increased incidence of immune-related AEs [31,40], no difference in immune-related AEs were seen across groups in this cohort. These data highlight the need to better understand the relationship between these inflammatory markers and the development of AEs, as well as the effects of the immune milieu on response to immunotherapy.

This is the first study investigating predictive value of NLR and PLR in patients with unresectable HCC treated with Atezo-Bev using real-world data from an international, multicenter cohort and has the advantage of being more broadly generalizable across diverse populations. However, our study also has several limitations, including those inherent to retrospective cohort studies. One of the major limitations of our study is the lack of NLR and PLR data across various time-points, which could have offered insight into how changes in the levels of these inflammatory markers could potentially inform response to therapy, patient outcomes, or development of AEs. Given the lack of records regarding other comorbidities, there is a possibility that the measurement of NLR and PLR could have been confounded by concomitant medications or other causes of chronic inflammation unrelated to HCC or liver disease. There also remains some controversy regarding the optimal cutoff value for NLR and PLR, but our study chose a more stringent cutoff to improve data accuracy. Importantly, patients with cirrhosis and portal hypertension are more likely to have thrombocytopenia, which would indicate more advanced liver disease and act as a confounder for measuring PLR. However, by only including patients with CP class A liver function, we reduced the impact of thrombocytopenia due to cirrhosis on PLR, and more directly, the median platelet counts of the two PLR groups in this study were nearly identical. Further, the evaluation of AEs was determined by chart review, which can be subject to incomplete documentation in the medical record during routine clinical practice. Finally, the value of NLR and PLR as predictive rather than prognostic biomarkers could not be established from this study without a control cohort of patients who did not receive Atezo-Bev.

## 5. Conclusions

In this multi-center, international cohort study, NLR ≥ 5 was associated with shorter OS in patients in unresectable HCC treated with Atezo-Bev. NLR is easily accessible, inexpensive, and practical in routine clinical practice, and it may be a prognostic marker worthy of further evaluation through prospective studies. The role of inflammatory biomarkers in the treatment of HCC using immunotherapy would also benefit from contextualization with other biomarkers, including markers of immune activation obtained from tissue specimens. As ICIs in combination with targeted therapies becomes the new treatment paradigm for unresectable HCC in the front line, more studies are needed to advance understanding of the effect of systemic inflammation on tumor progression and response to therapy.

## Figures and Tables

**Figure 1 cancers-14-05834-f001:**
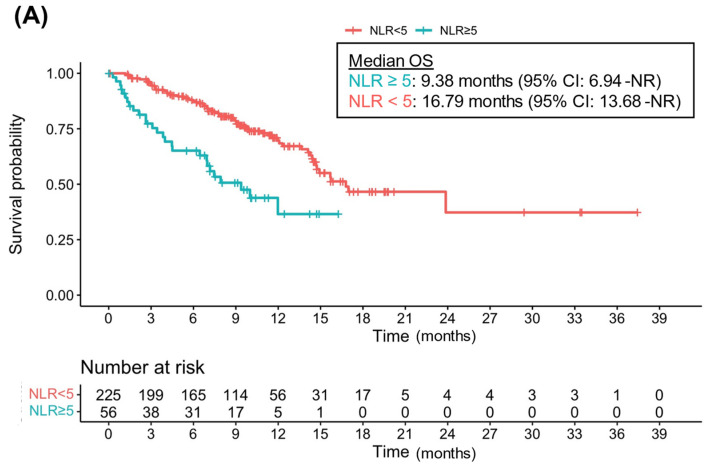
Kaplan–Meier curves for overall survival (OS) according to neutrophil-to-lymphocyte ratio (NLR) and platelet-to-lymphocyte ratio (PLR) status: (**A**) OS according to NLR; (**B**) OS according to PLR.

**Figure 2 cancers-14-05834-f002:**
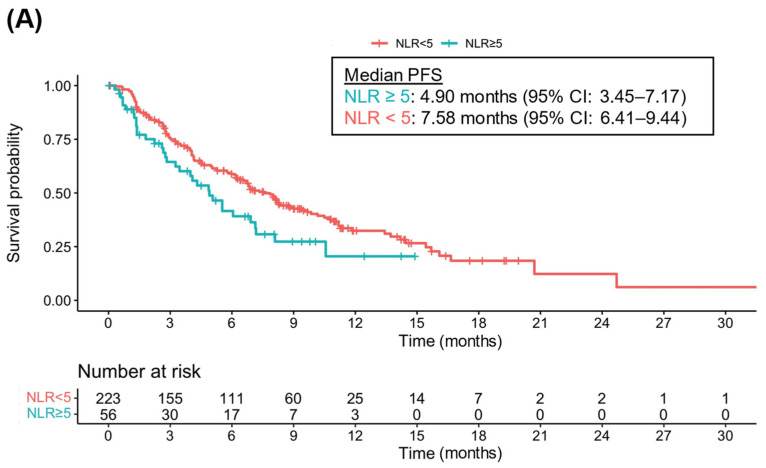
Kaplan–Meier curves for progression-free survival (PFS) according to neutrophil-to-lymphocyte ratio (NLR) and platelet-to-lymphocyte ratio (PLR) status: (**A**) PFS according to NLR; (**B**) PFS according to PLR.

**Figure 3 cancers-14-05834-f003:**
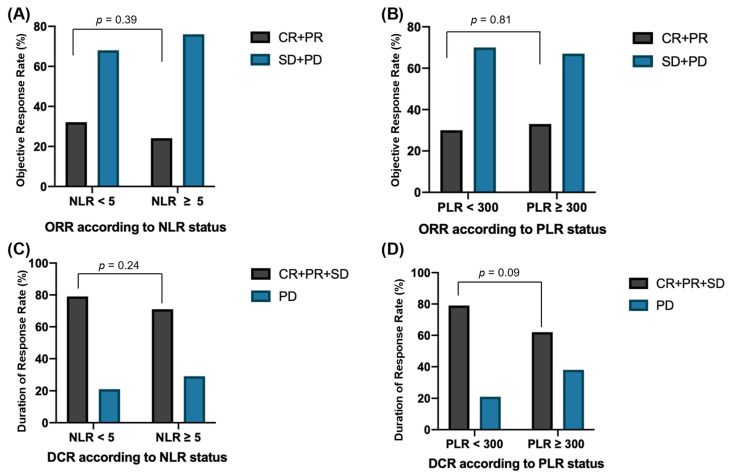
Bar graphs demonstrating objective response rate (ORR) and disease control rate (DCR) according to neutrophil-to-lymphocyte ratio (NLR) and platelet-to-Lymphocyte Ratio (PLR) status: (**A**) ORR according to NLR; (**B**) ORR according to PLR; (**C**) DCR according to NLR; (**D**) DCR according to PLR.

**Table 1 cancers-14-05834-t001:** Baseline Characteristics.

Whole Cohort *N* = 296			NLR ≥5 *N* = 56	NLR <5 *N* = 225	*p* Value	PLR ≥300 *N* = 24	PLR <300 *N* = 257	*p* Value
Age Median (IQR)		66 (59–73)	67 (61–74)	65 (57–63)	0.31	63 (52–71)	66 (59–73)	0.63
Male Gender *N* (%)		245 (82.7%)	47 (83.9%)	186 (82.7%)	0.82	19 (79.1%)	214 (83.3%)	0.58
ECOG PS		139 (47.0%)	21 (38.8%)	109 (48.9%)	0.18	9 (37.5%)	121 (47.1%)	0.40
*N* (%)	0							
	1	157 (53.0%)	35 (61.2%)	115 (51.1%)		15 (62.5%)	135 (52.9%)	
BCLC Stage		92 (31.3%)	14 (25%)	75 (33.3%)	0.26	5 (20.9%)	84 (32.7%)	0.26
*N* (%)	B							
	C	204 (68.9%)	42 (75%)	150 (66.7%)		19 (79.1%)	173 (67.3%)	
Extrahepatic Spread *N* (%)		169 (51.7%)	26 (48.1%)	115 (51.1%)	0.78	11 (45.8%)	125 (48.6%)	0.83
Cirrhosis *N* (%)		222 (75.0%)	42 (75.0%)	169 (75.1%)	0.99	15 (62.5%)	163 (63.4%)	0.99
Viral Etiology *N* (%)		195 (65.9%)	38 (67.8%)	151 (67.1%)	0.91	19 (79.1%)	170 (66.1%)	0.25
PVT *N* (%)		104 (35.0%)	28 (51.9%)	74 (32.9%)	0.020 *	11 (45.8%)	91 (35.4%)	0.38
AFP Median (IQR)		70 (6.5–1525)	268 (14–4211)	62 (6.42–1393)	0.02 *	608 (32–6331)	66.8 (6.50–1420)	0.08
Child Pugh Score		190 (64.2%)	29 (51.9%)	153 (68%)	0.029 *	13 (54.2%)	169 (65.8%)	0.27
*N* (%)	5							
	6	106 (35.8%)	26 (48.1%)	72 (32%)		11 (45.8%)	88 (34.2%)	
ALBI Grade		161 (54.4%)	18 (32.1%)	137 (60.1%)	<0.001 *	10 (41.7%)	145 (56.4%)	0.20
*N* (%)	1							
	2	135 (45.6%)	38 (67.9%)	88 (39.9%)		14 (58.3%)	112 (43.6%)	
Previous LRT *N* (%)		186 (59.6%)	33 (58.9%)	144 (64%)	0.53	15 (62.5%)	162 (63.0%)	0.96

* Statistically significant (*p* value < 0.05). NLR: neutrophil-to-lymphocyte ratio; PLR: platelet-to-lymphocyte ratio; IQR: interquartile range; ECOG PS: Eastern Cooperative Oncology Group performance status; BCLC: Barcelona Clinic Liver Cancer; PVT: portal vein thrombosis; AFP: alpha fetoprotein; ALBI: albumin-bilirubin; LRT: locoregional therapy.

**Table 2 cancers-14-05834-t002:** Univariate and Multivariate Cox Regression Analyses for Overall Survival.

	Univariate		Multivariate	
	HR; 95% CI	*p* Value	HR; 95% CI	*p* Value
Age	1.00; 0.98–1.02	0.80		
Gender (Male vs. Female)	0.89; 0.53–1.51	0.70		
BMI	1.00; 0.99–1.00	0.10		
BCLC C vs. B	1.42; 0.90–2.23	0.10		
AFP ≥400 vs. <400	1.72; 1.15–2.59	0.009 *	1.43; 0.93–2.20	0.10
ALBI 2 vs. 1	3.65; 2.36–5.64	<0.001 *	2.35; 1.42–3.89	<0.001 *
Child Pugh 6 vs. 5	2.42; 1.24–4.05	<0.001 *	1.38; 0.87–2.17	0.17
NLR ≥5 vs. <5	2.71; 1.71–4.27	<0.001 *	2.01; 1.22–3.56	0.007 *
PLR ≥300 vs. <300	2.24; 1.71–4.27	0.007 *	1.01; 0.52–1.96	0.99
Cirrhosis Yes vs. No	1.21; 0.74–1.99	0.40		
PVT Yes vs. No	2.03; 1.39–2.99	<0.001 *	1.52; 0.99–2.33	0.06
Extrahepatic Yes vs. No	0.93; 0.62–1.40	0.70		
ECOG PS 1 vs. 0	1.26; 0.83–1.90	0.30		
Viral vs. Non-Viral Etiology	0.95; 0.62–1.5	0.80		
Previous LRT Yes vs. No	0.52; 0.35–0.79	0.002 *	0.69; 0.44–1.08	0.10

* Statistically significant (*p* value < 0.05). HR: hazard ratio; CI: confidence interval; BMI: body mass index; BCLC: Barcelona Clinic Liver Cancer; AFP: alpha fetoprotein; ALBI: albumin-bilirubin; NLR: neutrophil-to-lymphocyte ratio; PLR: platelet-to-lymphocyte ratio; PVT: portal vein thrombosis; ECOG PS: Eastern Cooperative Oncology Group performance status; LRT: locoregional therapy.

**Table 3 cancers-14-05834-t003:** Univariate and Multivariate Cox Regression Analyses for Progression-Free Survival.

	Univariate		Multivariate	
	HR; 95% CI	*p* Value	HR; 95% CI	*p* Value
Age	0.99; 0.99–1.01	0.90		
Gender (Male vs. Female)	0.79; 0.53–1.16	0.20		
BMI	1.00; 0.99–1.00	0.10		
BCLC C vs. B	1.29; 0.94–1.79	0.10		
AFP ≥400 vs. <400	1.51; 1.11–2.05	0.009 *	1.41; 1.02–1.93	0.035 *
ALBI 2 vs. 1	1.57; 1.16-2.2	0.003 *	1.40; 1.03–1.92	0.034 *
Child Pugh 6 vs. 5	1.271; 0.93–1.74	0.10		
NLR ≥5 vs. <5	1.54; 1.05–2.25	0.03 *	1.31; 0.84–2.04	0.24
PLR ≥300 vs. <300	1.72; 1.04–2.83	0.04 *	1.18; 0.65–2.13	0.59
Cirrhosis Yes vs. No	0.97; 0.68–1.37	0.80		
PVT Yes vs. No	1.25; 0.93–1.68	0.10		
Extrahepatic Yes vs. No	0.99; 0.99–1.01	0.90		
ECOG PS 1 vs. 0	0.79; 0.53–1.16	0.20		
Viral vs. Non-Viral Etiology	1.23; 0.89–1.70	0.20		
Previous LRT Yes vs. No	0.74; 0.53–1.04	0.09		

* Statistically significant (*p* value < 0.05). HR: hazard ratio; CI: confidence interval; BMI: body mass index; BCLC: Barcelona Clinic Liver Cancer; AFP: alpha fetoprotein; ALBI: albumin-bilirubin; NLR: neutrophil-to-lymphocyte ratio; PLR: platelet-to-lymphocyte ratio; PVT: portal vein thrombosis; ECOG PS: Eastern Cooperative Oncology Group performance status; LRT: locoregional therapy.

**Table 4 cancers-14-05834-t004:** Incidence of Treatment-Related Adverse Events According to NLR and PLR Status.

	Any Grade TRAEs	*p* Value	Grade 3 or Higher TRAEs	*p* Value	irAEs	*p* Value
Whole Cohort	221 (74.7%)		70 (23.6%)		63 (21.3%)	
NLR ≥ 5	66.1%	0.08	14.8%	0.04 *	22.2%	0.96
NLR < 5	77.8%		27.5%		21.8%	
PLR ≥ 300	58.8%	0.05	16.7%	0.46	20.8%	0.91
PLR < 300	77.1%		25.7%		21.8%	

* Statistically significant (*p* value < 0.05). NLR: neutrophil-to-lymphocyte ratio; PLR: platelet-to-lymphocyte ratio; TRAE: treatment-related adverse event; irAE: immune-related adverse events.

## Data Availability

The data supporting the findings of this study are available upon reasonable request.

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
