# Peer review of "Neutrophil-to-Lymphocyte and Platelet-to-Lymphocyte Ratios as Prognostic Biomarkers in Unresectable Hepatocellular Carcinoma Treated with Atezolizumab plus Bevacizumab"

_cancers, 2022, doi:10.3390/cancers14235834_

Round 1

Reviewer 1 Report

Authors in this manuscript have aimed to identify the prognostic value of NLR and PLR in a real-world cohort of patients with unresectable HCC treated with Atezolizumab and Bevacizumab. Overall survival (OS), progression-free survival (PFS), Response to Atezo-Bev therapy and the incidence of AEs with Atezo-Bev and the association with NLR and PLR were assessed.  Although there are several limitations in the current study that affect the quality of manuscript. Overall the data are important for clinicians in the field. The first and most important weak aspect is correlated with the originally/novelty, once this work did not bring a significant innovation, because previous studies have been similar aims, such as (doi:10.3390/cancers13112786, doi:10.1002/cam4.3135 and doi:10.3390/cancers14020343). Thus, although this work studied a specific cohort (uHCC patients treated with atezolizumab plus bevacizumab), some aims are similar to those published previously. But this is the first study using real-world data from an international, multicenter cohort and has the advantage of being more broadly generalizable across diverse populations. Considering the number of patients treated with Atezo-Bev in the first line, I recommend this article for Cancers journal.

Author Response

We would like to thank the reviewer for a thoughtful and considered assessment of our manuscript. We agree that the idea to evaluate NLR and PLR as markers in HCC is not itself novel, but given the recent advances in the treatment of HCC, we believe that understanding the performance of these markers in the context of atezolizumab plus bevacizumab is of interest to those in the field. One of the strengths of our manuscript is its generalizability across international populations. We appreciate the reviewer's recommendation for publication in the journal. 

Reviewer 2 Report

The manuscript described a multicenter, international study on the prognostic value of NLR and PLR on HCC treated with Atezo-Bev. The study is similar to the study published by Jing-Houng Wang group on Cancers in January 2022. However, the manuscript included a larger cohort and involved patients from many centers internationally. Therefore, the conclusion is more generalizable. In additional, the authors also investigated responses and adverse effects between NLR/PLR high group and low group. 

The manuscript is well-written and easy to follow. Therefore, my overall suggestion is to accept in present form.

Author Response

We would like to thank the reviewer for a thoughtful reading of the manuscript. Although other groups have indeed evaluated the prognostic value of NLR and PLR, our study is bolstered by our international cohort representing diverse populations and additional analyses. We appreciate the reviewer's recommendation for acceptance of our manuscript.

Reviewer 3 Report

Nov. 18, 2022

Review comments to cancers-2038166

Wu YL, et al. collected the clinical materials from 296 patients with unresectable HCC and analyzed the prognostic values of NLR and PLR in those patients who were treated with Atezolizumab and bevacizumab. The authors assessed the association of NLR and PLR with OS, PFS and ORR. This is a very interesting, well-designed, well-written, and relevant clinical research manuscript. The results are very encouraging and useful for the clinicians. There are a few minor issues as below:

Major issues:

1.     Whether “response rate” in line 69 should be “objective response rate”?

2.     It is better to add “significant” between a and shorter in line 70 as the p < 0.001. Likewise, a “significant” can be added before decreased OS in line 73 and the p = 0.007.

3.     It is better to add (month) after the label “time” on X axis in both Fig. 2A and Fig. 1B.

4.     It is better to add (month) after the label “time” on X axis in both Fig. 2A and Fig. 1B.

5.     Labels on the Y axes are missing in Fig. 3 panel. % Objective Response Rate (ORR) should be added on Y axis in both Fig. 3A and Fig. 3B. Likewise, % Disease Control Rate (DCR) should be added on Y axis in both Fig. 3C and Fig. 3D.

Author Response

We would like to thank the reviewer for a careful reading of the manuscript. We have made the following corrections as suggested:

1. Whether “response rate” in line 69 should be “objective response rate”?

We agree that "objective" would be clearer. It has been added.

2. It is better to add “significant” between a and shorter in line 70 as the p < 0.001. Likewise, a “significant” can be added before decreased OS in line 73 and the p = 0.007.

"Significantly" was added in both lines. 

3. It is better to add (month) after the label “time” on X axis in both Fig. 2A and Fig. 1B.

We agree. The "months" label was added to the X axes.

4. It is better to add (month) after the label “time” on X axis in both Fig. 2A and Fig. 1B.

The "months" label was added to the X axes.

5. Labels on the Y axes are missing in Fig. 3 panel. % Objective Response Rate (ORR) should be added on Y axis in both Fig. 3A and Fig. 3B. Likewise, % Disease Control Rate (DCR) should be added on Y axis in both Fig. 3C and Fig. 3D.

The correct labels were added to the Y axes.